

# Putative archaeal viruses from the mesopelagic ocean

Dean R. Vik[1], Simon Roux[1], Jennifer R. Brum[1], Ben Bolduc[1], Joanne B. Emerson[1], Cory C. Padilla[2], Frank J. Stewart[2] and Matthew B. Sullivan[1,3]

[1] Department of Microbiology, Ohio State University, Columbus, OH, United States of America
[2] Department of Biological Sciences, Georgia Institute of Technology, Atlanta, GA, United States of America
[3] Department of Civil, Environmental and Geodetic Engineering, Ohio State University, Columbus, OH, United States of America

Corresponding author
Dean R. Vik, vik.1@osu.edu

## ABSTRACT

Oceanic viruses that infect bacteria, or phages, are known to modulate host diversity, metabolisms, and biogeochemical cycling, while the viruses that infect marine Archaea remain understudied despite the critical ecosystem roles played by their hosts. Here we introduce "MArVD", for Metagenomic Archaeal Virus Detector, an annotation tool designed to identify putative archaeal virus contigs in metagenomic datasets. MArVD is made publicly available through the online iVirus analytical platform. Benchmarking analysis of MArVD showed it to be >99% accurate and 100% sensitive in identifying the 127 known archaeal viruses among the 12,499 viruses in the VirSorter curated dataset. Application of MArVD to 10 viral metagenomes from two depth profiles in the Eastern Tropical North Pacific (ETNP) oxygen minimum zone revealed 43 new putative archaeal virus genomes and large genome fragments ranging in size from 10 to 31 kb. Network-based classifications, which were consistent with marker gene phylogenies where available, suggested that these putative archaeal virus contigs represented six novel candidate genera. Ecological analyses, via fragment recruitment and ordination, revealed that the diversity and relative abundances of these putative archaeal viruses were correlated with oxygen concentration and temperature along two OMZ-spanning depth profiles, presumably due to structuring of the host Archaea community. Peak viral diversity and abundances were found in surface waters, where *Thermoplasmata* 16S rRNA genes are prevalent, suggesting these archaea as hosts in the surface habitats. Together these findings provide a baseline for identifying archaeal viruses in sequence datasets, and an initial picture of the ecology of such viruses in non-extreme environments.

## INTRODUCTION

Viruses that infect bacteria, or phages, are relatively well-studied in the oceans and are thought to infect approximately one-third of seawater microbes at any given time (reviewed in *Brum & Sullivan, 2015*). Consequently, phage infections can have large impacts on marine ecosystems and microbial evolution in a variety of ways. First, phages alter microbial community structure and ecosystem functioning through host cell lysis.

Though difficult to quantify, this is thought to significantly increase the DOM pool, spur microbial growth (*Middelboe, Jørgensen & Kroer, 1996*; *Fuhrman, 1999*; *Wilhelm & Suttle, 1999*) and could help form cell debris aggregates that sink out of the water column, contributing to carbon export to the deep ocean (*Guidi et al., 2016*). Second, some phages hijack their host's metabolic machinery (*Vos et al., 2009*), reviewed in *Hurwitz & U'Ren (2016)*, which alters host cell metabolite concentrations (*Ankrah et al., 2014*; *Dendooven et al., 2016*), and can have direct impacts on ecosystem critical microbial metabolisms including carbon fixation, redox potential, nitrogen and sulfur cycling, and archaeal ammonia oxidation (*Hurwitz, Hallam & Sullivan, 2013*; *Puxty et al., 2016*; *Thompson et al., 2011*; *Lindell et al., 2005*; *Roux et al., 2016*; *Anantharaman et al., 2014*). Third, phages act as vectors for horizontal gene transfer among susceptible hosts with *bona fide* gene transfer now demonstrated in cyanophages for photosynthesis genes, and tentatively for genes implicated in deoxythymidine monophosphate production and the reduction of cellular guanosine pentaphosphate (*Lindell et al., 2004*; *Millard et al., 2004*; *Sullivan et al., 2006*; *Ignacio-Espinoza & Sullivan, 2012*; *Bryan et al., 2008*). Finally, the evolutionary trajectories of phages and their hosts are intimately connected via an arms race for infection or resistance mechanisms (*Comeau & Krisch, 2005*; *Stern & Sorek, 2011*; *Stoddard, Martiny & Marston, 2007*; *Koskella & Brockhurst, 2014*).

In contrast, most of the knowledge regarding the viruses infecting Archaea stems from extreme environmental isolates or enrichments, leaving archaeal viruses in the oceans understudied. To date, only two marine archaeal viruses are in cultivation, both of which were isolated from hydrothermal vents using thermophilic hosts: PAV1 isolated from *Pyrococcus abyssi* (*Geslin et al., 2003*), and TPV1 isolated from *Thermococcus prieurii* (*Gorlas et al., 2012*; *Gorlas et al., 2013*). Archaeal virus-related contigs have also been observed in a marine *Thermococcales* plasmid (*Soler et al., 2010*; *Zivanovic et al., 2009*) and in a genome from an anaerobic sediment viral metagenome (virome) in a marine methane seep (*Paul et al., 2015*). The latter, ANMV-1, is thought to infect Archaea because it contains TATA-box binding proteins specific to Archaea and Eukarya as well as six other genes (of 69 in the genome) that are similar to those from methanotrophic Archaea (*Paul et al., 2015*). Another putative *Thaumarchaeota* virus (Oxic1_7) was found in a fosmid library from Saanich Inlet in British Columbia (*Chow et al., (2015)*. Only one verified archaeal virus (AAA160-J20) has been identified from mesophilic ocean waters, recovered in a single-cell amplified genome from a *Thaumarchaeota* (*Labonté et al., 2015*). However recently, analysis of the "global ocean virome" has revealed three additional candidate viral genera putatively associated with Archaea according to matching CRISPR spacers, similar tetra-nucleotide frequencies and BLASTn homology to reference sequences (*Roux et al., 2016*). In spite of the fact that Archaea often comprise 5–40% of the marine microbial community and play critical ecological roles (e.g., nitrogen cycling), the viruses described above are the only representation of archaeal viruses in the marine realm to date. Given the high abundance of Archaea in the oceans (*Ganesh et al., 2015*; *Karner, DeLong & Karl, 2001*), this suggests that there may be vast archaeal virosphere yet to be explored if viruses play as large a role in the ecology of Archaea as they do for Bacteria (*Danovaro et al., 2016*).

Here we introduce MArVD (Metagenomic archaeal virus detector), an annotation mining tool to automate the identification of putative archaeal virus contigs in viral metagenomic datasets from marine samples. We applied MarVD to identify archaeal viruses in 10 marine viromes from an oxygen minimum zone (OMZ) in the Eastern Tropical North Pacific (ETNP). Strong OMZs are associated with upwelling zones of the Eastern Pacific, including the ETNP, due to heterotrophic respiration of surface primary productivity in waters below the photic zone. In the ETNP, oxygen concentrations fall below detection (<10 nM) (*Ulloa et al., 2012*; *Tiano, Garcia-Robledo & Revsbech, 2014*) and chemolithoautotrophs and heterotrophs, including diverse assemblages of Archaea, adapt to such low oxygen by using anaerobic metabolisms that impact geochemical cycling globally (*Canfield, Glazer & Falkowski, 2010*; *Lam & Kuypers, 2011*; *Wright, Konwar & Hallam, 2012*; *Kalvelage et al., 2015*). Although ammonia-oxidizing *Thaumarchaeota*, as well as diverse members of the *Euryarchaeota*, are abundant above or within the anoxic OMZs of the Eastern Pacific (*Stevens & Ulloa, 2008*; *Stewart, Ulloa & Delong, 2012*; *Wright, Konwar & Hallam, 2012*), the viral diversity in these systems has been largely unexplored apart from a single study, which revealed novel viral assemblages, of unknown host, unique to the anoxic core of the Eastern Pacific OMZ near Chile (*Cassman et al., 2012*). By applying MArVD to 10 OMZ viromes, 43 genomes and genome fragments, putatively associated with archaeal hosts, were detected and analyzed in an ecological context to develop the first tentative population-based picture of mesophilic, marine archaeal virus ecology in a climate-critical OMZ containing marine habitat.

## MATERIALS AND METHODS

### Virome sample collection

Ten sea water samples were collected on June 13–28 2013 on the R/V New Horizon at two locations in the ETNP off the west coast of Mexico (Station 2, 18°55′12″N and 108°47′60″: station 6, 18°55′12″N and 104°53′24″W). Station 6 is positioned near shore over the continental shelf while station 2 is roughly 450 km directly west of station 6 in the open ocean. At each station, 20 liters of water were collected using Niskin bottles attached to a rosette. Sample depths were 30 m, 60 m, 130 m, 300 m, and 1,000 m at station 2 and 30 m, 85 m, 100 m, 300 m, and 1,000 m at station 6, which represented samples from the surface mixed layer, upper oxycline, OMZ core and lower oxycline at either station respectively. Measurements of oxygen concentration, temperature, salinity, and chlorophyll *a* were recorded for each sampling effort using a Conductivity Temperature Depth profiler (Sea-Bird SBE 911plus, Sea-Bird Electronics Inc., Bellevue, WA, USA) along with a Seapoint fluorometer (Seapoint Sensors Inc., Exeter, NH, USA) and a SBE43 dissolved oxygen sensor (Sea-Bird Electronics Inc.) (Data S5).

### Virome preparation

Each of the 20-liter water samples was passed through a 0.22 um filter on a 142 mm diameter Express Plus filter apparatus to remove prokaryotic cells and cellular debris. The viral fraction of the filtrate was then concentrated using iron chloride to flocculate the viral particles, which were then stored at 4 °C indefinitely. Resuspension of the viral particles

was done using an ascorbic EDTA buffer (0.1 M EDTA, 0.2 M Mg, 0.2 M ascorbic acid, pH 6.0) to facilitate further concentration of the viral particles using an Amicon Ultra 100-kd centrifuge (Millipore inc.). The viral concentrates were then treated with DNase I, and 0.1 M EDTA and EGTA to eliminate any free DNA from the samples. Viral DNA was extracted using the Wizard prep PCR purification kit with 0.5 ml sample added to the 1ml resin and eluted on Wizard mini-columns with TE buffer (10 mM Tris, pH 7.5, 1 mM EDTA) (Promega, Fitchburg, WI, USA). The extracted DNA was then sheared with a Covaris ultra-sonicator, and gel purified to select fragments of 160–180 bp in length. The final sequencing was carried out on a HiSEq 2,000 system at the DOE Joint Genome Institute. Reads were then quality trimmed to remove bases with quality scores greater than two standard deviations from the average score (across sequencing cycles), and bases with a quality score lower than 20. Lastly, a size threshold of 95 bp was imposed on the entire read dataset. Reads were assembled using the SOAPdenovo software in the MOCAT pipeline. The assembled contigs were then confirmed to be viral in origin using the VirSorter software with default settings (*Roux et al., 2015a*; *Roux et al., 2015b*). The viral fraction of the ETNP metagenomic dataset was uploaded to MetaVir 2 (*Roux et al., 2014*) and those contigs larger than 1,000 bp and with three or more ORFs, at least one of which affiliated to a reference archaeal virus or archaeal virus pathway, were considered for further analysis. In order to focus on only those viruses which meet the requirements to be identified as populations according to recently established metrics for this classification, (*Brum & Sullivan, 2015*) these sequences were further filtered using the Cdhit software to retain only contigs greater than 10,000 bp in size and <95% average nucleotide identity over 80% of the shorter sequence (*Fu et al., 2012*).

## MArVD

In order to identify marine archaeal viruses in metagenomic datasets, we created three categories of archaeal virus-like sequences based on the proportion of genes per contig that affiliate with reference archaeal viruses in the RefSeq database (version 74), and the score of these assignments according to a BLASTp analysis. The first category collects contigs in which more than 66.6% of the genes per contig are annotated as archaeal virus, the bit-score of at least one of these annotations is higher than 75, and the bit-score exceeds those given to all genes with phage annotations. The second category again collects contigs in which more than 66.6% of the genes affiliated with a reference archaeal viruses and at least one of these annotations has a bit-score higher than 75, but in this instance, at least one gene assigned as phage may have a bit-score higher than that for genes assigned as archaeal virus. The final category includes contigs with 33.3–66.6% of genes annotated as archaeal virus, at least one of these annotations has a bit-score higher than 75, and again at least one of these annotations has a bit-score exceeding phages annotations. Eukaryotic viruses were not considered in these categories.

Each of these categories were then implemented in a python script which we named MArVD. MArVD assigns a contig to a specified category by using an input file in the form of a detailed gene-per-contig annotation file, which is output from the Metavir2 software (*Roux et al., 2014*). MArVD then uses a keyword searching methodology in

conjunction with an user-tailored input file containing a keyword list, to identify taxonomic affiliations, linking genes to either reference archaeal viruses, or Archaea. It then identifies the proportion of archaeal virus genes in each contig and the quality of the taxonomic assignment in bit-score and parses the annotation table, separating contigs which either meet or fall short of the established thresholds for a designated category. The gene proportion and annotation quality threshold in MArVD are modifiable options in order to allow the user to control the categorical thresholds if desired.

We then determined the accuracy and sensitivity of MArVD considering all categories collectively, and for each individual category, with the published Virsorter dataset with viruses of known host.

**Putative archaeal virus contig network analysis**

To further analyze the genetic relatedness between the 43 ETNP archaeal viruses populations we used a Markov Clustering Algorithm (MCL) based network analysis as previously described in *Lima-Mendez et al. (2008)*. For this analysis reference phage and archaeal virus genomes from RefSeq (version 74) were combined with the ETNP archaeal virus contigs in order to contextualize them relative to other known viruses (*Finn et al., 2014*; *Bolduc et al., 2016*). This algorithm first groups ORFs using an MCL clustering algorithm based on BLASTp $e$-values, and the shared gene content between each contig is then used to further group contigs into viral clusters or VCs, which were then represented as a network and visualized in Cytoscape version 3.2.1 using the ''Edge Weighted Spring Embedded Layout'' (*Shannon et al., 2003*). Viral clusters were then organized according to their predicted host from our previous analysis or reference information.

**43 putative archaeal virus genomic comparison**

Easyfig version 2.2.2 was used to draw contig comparison maps, with similarities detected using GenBank files for each of the 43 archaeal virus populations via tBLASTx analysis and an $e$-value threshold of 0.001. The resulting images were tuned to display the correct organization and orientation of each sequence according to their similarity to the rest of the populations in each VC and centered around the best blast hit across each VC. Each gene's taxonomic affiliation regarding archaeal virus, phage or eukaryotic virus and its functional annotation were then reported. Where synteny and annotation were ambiguous based on best BLASTp affiliation, the taxonomy and function of the genes were inferred using a consensus among the top hits within 20% of the blast hit with the lowest $e$-value.

**Portal protein phylogenetic analysis**

Of the 43 archaeal virus populations, 17 encode the PF04860.7 portal protein. Sequences from all known viral variants of this gene were downloaded from the Pfam database (version 29.0) in order to conduct a single gene phylogenetic analysis (*Finn et al., 2014*). Sequences were aligned locally using MAFFT (version 7.222) and default parameters (*Katoh & Standley, 2013*). The resulting alignment file was manually curated with Jalview (version 2) to remove leading and tailing portions of the alignment which displayed low conservation, and to remove contigs which were obviously misaligned to limit the potential for long branch attraction (*Waterhouse et al., 2009*). The tree was constructed in

the FastTree (version 2.1) program using the nearest neighbor interchange and minimum evolution algorithm to develop an approximate maximum likelihood estimation based phylogenetic tree with the Jones-Taylor-Thorton evolutionary model, re-sampling 1,000 times for robust bootstrap values (*Price, Dehal & Arkin, 2010*). The final tree was visualized using the Interactive Tree of Life software (ITOL) (version 3) and sample location, depth, taxonomic affiliation, and viral cluster affiliation were then added as additional data layers (Fig. S2) (*Letunic & Bork, 2011*).

## Putative archaeal virus community analysis

Quality trimmed reads from the ETNP dataset were mapped to the 43 archaeal virus populations in order to determine the abundance and distribution of these sequences within the ETNP at both stations. Reads were first quality trimmed with the Trimmomatic software and mapped to their respective contigs using the Bowtie2 software (*Bolger, Lohse & Usadel, 2014*; *Langmead & Salzberg, 2012*). Coverage was calculated with the BamM package (*Li, 2011*; *Rabosky et al., 2014*). The resulting per base-pair coverage matrix displays the relative abundance and distribution of each viral sequence and can be used to infer the environmental factors influencing the distribution of the ETNP archaeal virus populations. Richness, Shannon-Wiener diversity index, and Pielou's Evenness values were calculated for each site either manually or using the vegan package for R studio (*Oksanen, 2015*; *Shannon & Weaver, 1949*). A heat-map highlighting patterns in the population distributions and abundances for each virome was then developed using the R studio package heatmap3 and its maximum likelihood clustering algorithm along with the Pvclust package (*Suzuki & Shimodaira, 2015*; *Kolde, 2015*). Colors were chosen using http://colorbrewer2.org. A non-metric multidimensional scaling analysis was performed in the vegan package with the Bray-Curtis distance algorithm using the read coverage matrix for both stations in order to group the communities found in each habitat. Amending this analysis with the extensive environmental data available for these locations allowed for the determination of which environmental factors most significantly influenced the observed archaeal virus population distribution. Bray–Curtis dissimilarity matrices from the 43 putative archaeal viruses abundance profiles were then compared via Mantel test using the R package Vegan.

## 16S rDNA extraction

Microbial community taxonomic composition was assessed via Illumina sequencing of 16S rRNA gene fragments amplified from bacterioplankton DNA from stations 2 and 6. Station 6 data were generated in a prior study (*Ganesh et al., 2015*) and reanalyzed here. Station 2 data were generated in this study, as follows. DNA was extracted from Sterivex filters (>0.2 $\mu$m biomass size fraction) from 5 depths spanning the oxic zone, oxycline, and OMZ using a phenol: chloroform protocol following protocols described in *Ganesh et al. (2015)*. Cells were lysed by adding lysozyme (2 mg in 40 $\mu$l of lysis buffer per filter) directly to the Sterivex cartridge, sealing the ends, and incubated for 45 min at 37 °C. Proteinase K (1 mg in 100 $\mu$l lysis buffer, with 100 $\mu$l 20% SDS) was added, and cartridges were resealed and incubated for 2 h at 55 °C. The lysate was removed, and the DNA was extracted once with

phenol:chloroform:isoamyl alcohol (25:24:1) and once with chloroform:isoamyl alcohol (24:1) and then concentrated by spin dialysis using Ultra-4 (100 kDa, Amicon) centrifugal filters.

## 16s rRNA gene library preparation

Sequences were generated using an established pipeline as in *Ganesh et al. (2015)* and *Padilla et al. (2015)*, *Padilla et al. (2016)*. Briefly, amplicons were synthesized using Platinum® PCR SuperMix (Life Technologies) with primers F515 and R806 encompassing the V4 region of the 16S rRNA gene (*Caporaso et al., 2011*). These primers are used primarily for bacterial 16S rRNA gene analysis, but also to amplify archaeal sequences. Both forward and reverse primers were barcoded and appended with Illumina-specific adapters according to *Kozich et al. (2013)*. Thermal cycling involved: denaturation at 94 °C (3 min), followed by 30 cycles of denaturation at 94 °C (45 s), primer annealing at 55 °C (45 s) and primer extension at 72 °C (90 s), followed by extension at 72 °C for 10 min. Amplicon size (∼400 bp, including barcodes and adaptor sequences) was verified by gel electrophoresis and amplicons were purified using Diffinity RapidTip2 PCR purification tips (Diffinity Genomics, NY). Amplicons from different samples were sequenced on an Illumina MiSeq using a 500-cycle kit.

## 16s rRNA gene analysis

Station 2 sequence data were combined with those generated previously for station 6 and analyzed using QIIME (*Caporaso et al., 2010*). Barcoded sequences were de-multiplexed and trimmed (length cutoff 100 bp) and filtered to remove low-quality reads (average Phred score <25) using Trim Galore! (http://www.bioinformatics.babraham.ac.uk/projects/trim_galore/). Paired-end reads were then merged using FLASH (*Magoč & Salzberg, 2011*), imposing a minimum average length of 250 bases for each read, a minimum average length of 300 bases for paired read fragments, and a maximum fragment standard deviation of 30 bases. Chimeric sequences were detected by reference-based searches using USEARCH (*Edgar, 2010*) and removed. Merged non-chimeric sequences were clustered into Operational Taxonomic Units (OTUs) at 97% sequence similarity using open-reference picking with the UCLUST algorithm (*Edgar, 2010*) in QIIME. The number of sequences following quality filtering ranged from 18,036 to 193,094 and the number of OTUs ranged from 527 to 1,023. Taxonomy was assigned to representative OTUs from each cluster using the Greengenes database (*DeSantis et al., 2006*). Total microbial (bacteria + archaea) and total archaeal OTU counts were determined based on 10 resampling iterations at uniform minimum sequence depths ($n = 18,036$ and 625 respectively), with the removal of any singleton OTUs or those associated with mitochondria and chloroplasts. Abundances for microbial Orders were calculated as proportions of total reads; only Orders comprising >0.5% of total reads (averaged across all samples) are displayed. Ordination of the 16s rRNA sequences from the archaeal fraction of this microbial community was analyzed in the same way as for the viral populations. Bray–Curtis dissimilarity matrices from the Archaea fraction of the 16 s rRNA data were then compared via Mantel test using the R package Vegan.
## RESULTS AND DISCUSSION

### MArVD: a tool to automatically and systematically identify putative archaeal virus contigs

To identify archaeal virus contigs in viromes, we developed the Metagenomic archaeal virus detector (MArVD) that identifies putative archaeal virus contigs based on a "majority rules" consensus taxonomic assignment drawn from the taxonomic annotation of each predicted gene (Fig. S1). MArVD entails three categories (detailed in the supplemental online material) used to filter archaeal virus contigs from metagenomic datasets. Briefly, the first category collects contigs with three or more per-gene taxonomic annotations, the majority of which (>66.6%) affiliate with reference archaeal viruses above a pre-defined quality threshold (bit-score 75), and with better quality then affiliations to reference phages. This category also includes contigs with just one or two taxonomic annotations that only affiliate with reference archaeal viruses. Because archaeal viruses are heavily under-represented in public datasets (*Dellas et al., 2014*), this first category accounts for the likelihood that most marine archaeal virus genes will be unclassified. The second category uses the same metrics as the first but allows for at least one gene to have a phage annotation of better quality than the archaeal virus annotations. This is intended to account for potential misclassification of archaeal virus genes as phage genes or the possibility that an archaeal virus has incorporated a phage gene via horizontal gene transfer (*Iverson et al., 2012*). The final category includes contigs with fewer (33.3–66.6%) archaeal virus annotations, but of bit-score quality exceeding the phage annotations. This category weights the taxonomic designation of contigs on the quality of gene annotation, rather than the proportion of genes affiliating with references. These categories were then implemented in a python script, which assigns a contig to a specified category by searching detailed gene annotation tables for hits to reference archaeal viruses (Fig. S1). Hits to Eukaryotic viruses were not considered because Eukaryotic viruses are likely rare in marine viromes (*Suttle, 2007*).

In order to determine the accuracy and sensitivity of MArVD's categories collectively and individually, we applied it to the published Virsorter dataset, which is a collection of viral contigs mined from publicly available microbial genomes, providing the host for each identified viral contig (*Roux et al., 2015a*; *Roux et al., 2015b*). MArVD was able to identify all 127 (100% sensitive) of the previously established archaeal viruses in this dataset and only miscategorized two of the 12,372 phage contigs (>99% accurate). Sensitivity and accuracy predictions per category revealed that category one identified 123 of the 127 archaeal viruses with only one of the two erroneously designated phage contigs. Category two identified one of the 127 archaeal viruses. Category three identified five contigs as archaeal viruses with one misidentified phage contig. The two phage contigs (5062_Planctomycetia_gi_163804221 and 5062_Gammaproteobacteria_gi_484067609_0_4307) contain only two (of 52 total) (category one) and three (of five total) (category three) ORFs that could be taxonomically annotated (*Roux et al., 2015a*; *Roux et al., 2015b*). While the rest of the ORFs in these contigs were unclassified, most of the taxonomic annotations available for both of these contigs affiliated with reference archaeal viruses, thus facilitating their misidentification.

While highly accurate on the Virsorter dataset, our stringent thresholds and reliance on existing databases mean that MArVD will not identify all archaeal viruses in more complex communities with novel viruses. MArVD is dependent upon annotation with pre-established reference databases and requires at least one gene to be taxonomically annotated for each contig. Because the ratio of total genes to annotated genes can be very low on short viral contigs, this pragmatically limits its usefulness to contigs >10 kb in size representing viral populations (*Brum et al., 2015*). To increase sensitivity where novel archaeal viruses are expected, future work might include an iterative approach whereby newly identified contigs can serve as references for a second (or more) application of MArVD.

## MArVD identifies 43 putative marine archaeal viruses in ETNP viromes

We next explored the marine archaeal virome as a baseline for hypothesis generation regarding marine archaeal virus ecology. To this end, we examined the waters of the ETNP OMZ, sampled across five depths at each of two stations (Fig. 1A) and spanning the strong oxygen gradients (Fig. 1B), where Archaea are common as inferred from 16S rRNA amplicon analyses. The relative proportion of total Archaea to Bacteria 16S sequences in these samples suggested that Archaea comprised 3–40% and 5–25% of the total microbial community in the surface and upper oxycline waters versus OMZ core waters, respectively (Fig. 1C). The dominant Archaea were from class *Thermoplasmata* and the phylum *Thaumarchaeota* (Fig. 1C). The abundant Archaea in the ETNP suggests an enrichment in archaeal viruses, so MArVD was applied to viromes derived from these same 10 samples (Data S1). While this approach undoubtedly under-represents the diversity of archaeal viruses in these waters, these putative archaeal virus genomes will provide a critically-needed first glimpse of their ecology.

MArVD identified 407 putative archaeal virus contigs, which were grouped according to current viral population-scale metrics (≥95% ANI, or average nucleotide identity, across 80% of the shorter contig) into 344 different viral 'populations' (Data S2A & S2B). Of these, only 43 contigs were ≥10 k bp in size (range = 10,051–31,425 bp; average = 14,852 bp) and considered long enough for further interrogation as putative archaeal virus populations (Data S1). One of these populations (2_300_C3851131) from 300m depth at the station farthest offshore (Station 2) was represented by a 12,262 bp circular contig, which, because it circularized may be complete, but if so is then shorter than the genome of any known *Caudovirales* isolate genome, suggesting circularization due to assembly artifacts. It contained 19 hypothetical genes and an archaeal virus-like terminase, as well as a portal protein, which was annotated as phage-like, but is phylogenetically divergent from known phage portal proteins (Fig. S2). The other 42 populations are non-circular and so are likely not complete.

## The 43 ETNP viruses represent six novel archaeal virus genera

Because viruses lack a universal marker gene, assigning taxonomy to new viruses can be challenging. To classify these 43 new putative archaeal viruses, we used a genome-
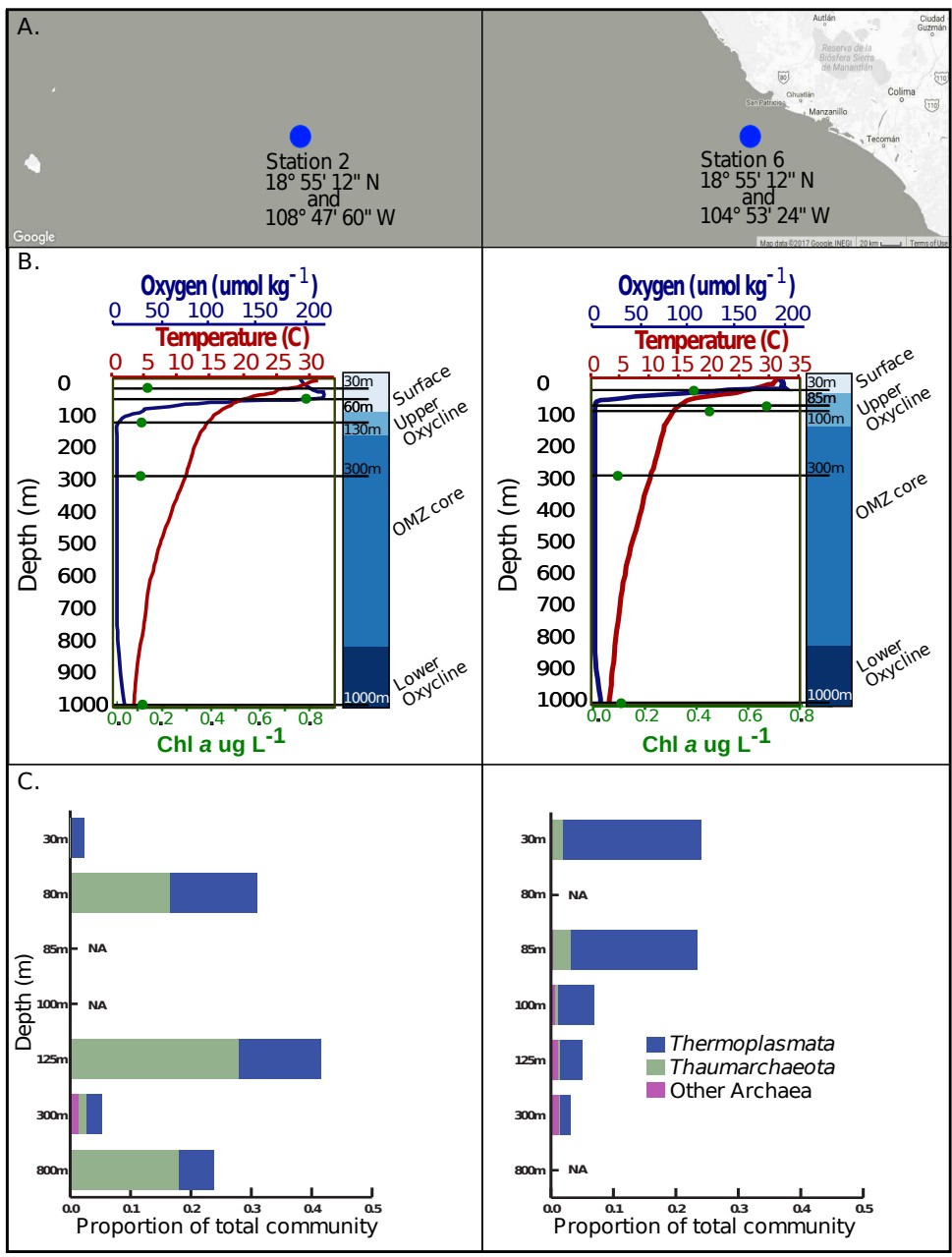

**Figure 1** **Sample locations, environmental features and Archaea 16s rRNA amplicon abundance profiles.** (A) Station 2 and station 6 are located west of Manzanillo Mexico, at the eastern edge of the Eastern Tropical North Pacific. (B) Five samples, denoted by the solid black lines corresponding to various depths, in the surface, upper oxycline, OMZ core, and lower oxycline habitats respectively were selected at each station. Both stations exhibit a sharp reduction in oxygen concentration, reaching a minimum of 1–3 umol kg$^{-1}$ at roughly 130 m in station 2 and 67 m in station 6. Both stations have a constant reductive thermocline ranging from about 28 °C in the surface to about 6 °C at 1,000 m. Chlorophyll *a* concentrations are highest in the surface waters and upper oxycline (0.4–0.8 ug/L) and are lowest in the OMZ core and below (∼0.1ug/L). (C) Archaeal 16s rRNA amplicon based relative abundance profiles indicate that *Thermoplasmata* and *Thaumarchaeota,* the predominant archaeal phyla present at these sites and have variable distributions on a depth gradient.

and network- based classification methodology previously used to group modules of protein clusters according to genetic similarity (*Lima-Mendez et al., 2008*; *Roux et al., 2015a*; *Roux et al., 2015b*; *Roux et al., 2016*; *Bolduc et al., 2016*). In these analyses, viral genomes are grouped based on their gene content profiles into viral clusters, or VCs, which are approximately equivalent to candidate viral genera as assigned by the International Committee on the Taxonomy of Viruses (edited by *King et al., 2012*). Our network analysis, conducted using "vConTACT" (*Bolduc et al., 2016*), included the 43 ETNP putative archaeal virus populations along with 1,573 phages, and 60 archaeal viruses from NCBI's RefSeq database (version 74). The resulting network revealed that the putative ETNP archaeal viruses, RefSeq archaeal viruses, and RefSeq phages each formed distinct modules. The only exception was a small VC of six reference genomes that included both phages and archaeal viruses, likely due to a 'long branch attraction' type grouping (*Wiens, 2005*), where highly divergent genomes are artificially placed together when they represent a poorly sampled region of sequence space (Fig. 2A, Data S3).

In this network, the putative ETNP archaeal viruses formed six entirely novel VCs, according to vConTACTs default settings, with no connections to any reference viral genomes (Fig. 2B). Comparative genomic analyses revealed that the members of the six ETNP VCs shared on average 58.9% of their genes within a VC and on average only 4.4% of their genes with any other VC in the entire network. VC-guided comparative genomic analyses also showed that while only 14.5% of the genes across all six ETNP VCs could be affiliated to a particular taxon, most (~74.8%) of these taxonomically annotated genes were most closely affiliated with known archaeal viruses, at high confidence (average bit-score per VC ranged from 109–223). The remaining ~25.2% of the taxonomically annotated genes affiliated with either phage (~13.9% of total annotations) or eukaryotic viruses (~11.3%), although the phage annotations were relatively weak (average bit-score per VC ranged from 68–111). The eukaryotic virus annotations were of similar quality as the archaeal virus annotations (average bit-score 216); however, these genes were only present in VC2, made up only 35% of annotated genes in this VC, and have an elevated average bit-score rating due to a single conserved gene found in four contigs (Figs. 3A–3F). The 25.2% of genes affiliated with phage or eukaryotic viruses are likely a reflection of database bias since mesophilic archaeal viruses are not currently well represented in public databases. Regardless, these analyses suggest that the 43 ETNP contigs likely represent novel mesophilic archaeal viruses due to the high proportion (74.8%) of hits to reference archaeal viruses, and since they form VCs distinct from reference phage.

Finally, we assessed the quality of our network-based classifications by comparison with a phylogenetic approach using a taxonomic marker gene that encodes the portal protein (Pfam family PF04860.7) involved in viral DNA packaging and transport. This gene was identified in 17 of the 43 putative archaeal virus populations, representing four of the six VCs (Star in Figs. 3A, 3C, 3E and 3F). The resulting portal protein phylogeny revealed a single monophyletic group for the putative ETNP archaeal viruses comprised of three main sub-clades, each corresponding to a single VC and including 15 contigs (Fig. S2). The single contig not placed in a clade with its corresponding VC is the only member of another subclade and shares two of its three edges in the network analysis with the

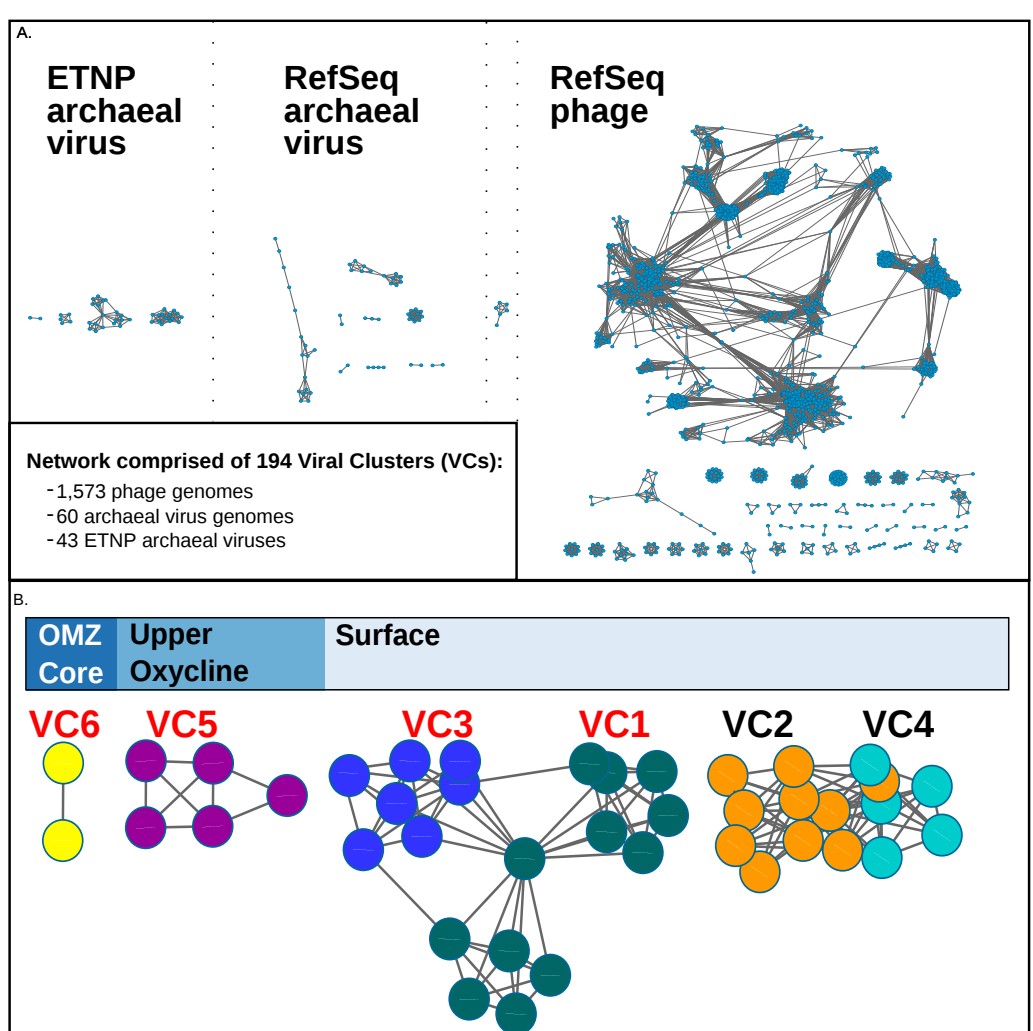

**Figure 2  vConTact Network analysis of reference of phage and archaeal virus sequences with the ETNP archaeal virus dataset.** (A) The vConTact network developed by using the entire RefSeq archaeal virus (60 sequences) and phage (1,573 sequences) database in conjunction with the ETNP dataset (43 sequences). (B) A focused view of the network comprised of the ETNP archaeal virus populations. The sample habitat at which each VC is found in the highest prevalence is displayed at the top of (B). VCs with red labels are represented in the phylogenetic analysis (Fig. S2).

most divergent member of the subclade corresponding to VC3, and a member of VC1 respectively. Thus, both means of classification are largely concordant and suggest that the putative ETNP archaeal viruses are highly divergent from other known phages and archaeal viruses. Additionally, using HMMscan searches against the Pfam database, we find that the portal protein analyzed here is represented in only 0.6% of the non-archaeal fraction of the ETNP viromic dataset, suggesting that this gene may help guide the identification of archaeal viruses in other marine environments (Data S4).

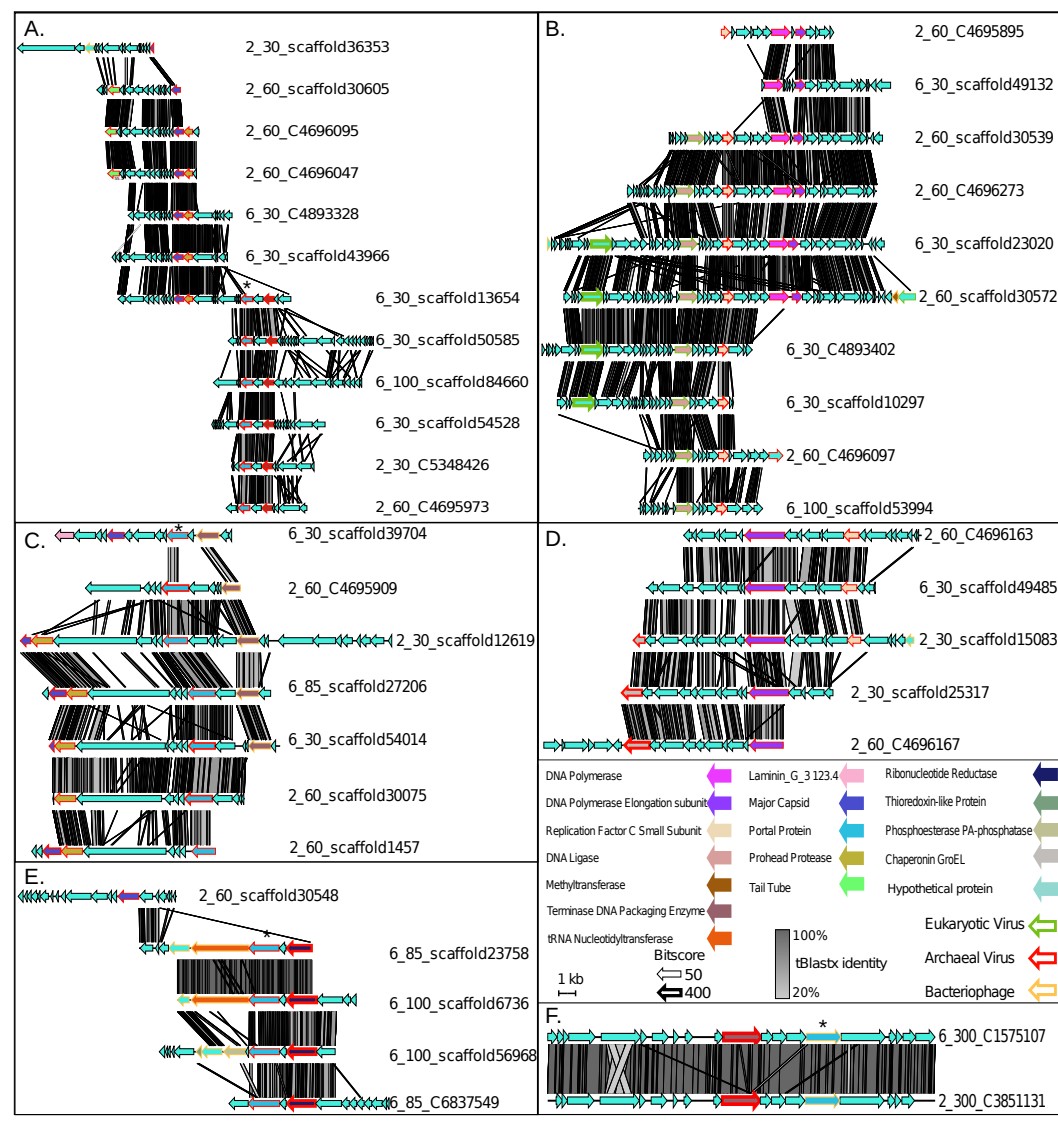

**Figure 3   Synteny maps for each of the putative ETNP archaeal virus cluster (VCs).** (A) VC1. (B) VC2. (C) VC3. (D) VC4. (E) VC5. (F) VC6. Each annotated gene's taxonomic affiliation is displayed as the colored outline with the bit-score value represented by the weight of the outline. Functional annotations are further depicted as the genes interior color. tBLASTx percent identities among related regions of each population are represented by the gray bar between each population. The portal protein used for verification of the network analysis via phylogenetics is denoted by a star (Fig. S2).

## Environmental factors influence archaeal virus population distributions in the ETNP

Given these 43 new putative archaeal virus genomes, we next explored their ecological patterns and drivers in the ETNP. Putative Archaeal virus abundance and diversity profiles were compared with the available meta-data including temperature, oxygen concentration, salinity, chlorophyll *a,* and depth (Data S5). Both the relative abundance and alpha diversity (Shannon–Wiener index) of archaeal viruses was highest in the surface waters and diminished along the gradient in oxygen concentration, reaching a low in the OMZ core.

The relative evenness (Pielous J) of the detected viral populations was consistently near 1 across the OMZ oxygen gradient, and above (Fig. 4A). Additionally, both sample stations displayed similar putative archaeal virus abundance and membership across all depths as revealed by comparison of Bray-Curtis dissimilarity matrices via mantel test ($p < 0.01$, mantels $r = 0.8375$) (Data S6).

Metagenomic read recruitment was used to estimate the abundance of these putative archaeal viruses at our two sampling sites. This revealed that the putative archaeal virus populations appeared to be stratified along the depth profile (Fig. 4B). This stratification was especially apparent at the interface between oxygenated and anoxic waters, across which no populations were shared (Data S6). This apparent niche differentiation is consistent with results from non-metric multi-dimensional scaling (NMDS) analysis, which revealed that both oxygen concentration ($p < 0.001$) and temperature ($p < 0.005$) were the two most significant factors driving separations in the viral communities (Figs. 5A and 5B).

Although further data is needed to disentangle the relative effects of temperature versus oxygen concentration on viral community structure, we posit that oxygen concentration is the more influential factor, inferred from the punctuated change in the viral community composition at the base of the oxycline (Fig. 4B). Neither salinity, the location of the deep chlorophyll maximum or station identity (2 vs. 6) were significant drivers of putative archaeal virus community structure in our analyses. This result is not surprising considering that there was minimal variation of salinity along either station's depth gradient, known Archaea do not contain chlorophyll (*Blankenship, 2010*) and the temperature and oxygen profiles were similar between these two stations.

Analysis of 16S rRNA genes in the ETNP via NMDS ordination revealed that the population distribution of Archaea in both stations is also significantly influenced by oxygen concentration ($p > 0.05$) (Fig. 5C) and temperature ($p < 0.005$) (Fig. 5D). This is consistent with the observed patterns in putative archaeal virus populations; however, temperature appeared to the most significant factor influencing the distribution of Archaea in these samples. Analysis of the 16s rRNA gene abundance and composition profiles suggests that a distinct archaeal community is found at both stations 2 and 6 (Fig. 1D) (Data S7).

The dissimilarity between the archaeal populations at either station was further verified by comparison of Bray–Curtis dissimilarity matrices via mantel test ($p > 0.5$, Mantels $r = -0.156$). Specifically, at station 6, *T hermoplasmata* were at the highest abundance and numerically dominated the archaeal community in the oxygenated surface waters, and *Thaumarchaeota* increased in abundance at the upper oxycline but displayed relatively low abundance in OMZ core and below (Fig. 1C). At station 2, *Thermoplasmata* is present at low abundances in the surface, became more abundant in the upper and lower oxycline, and were at low abundances in the OMZ core. The *Thaumarchaeota* in station 2 are absent in the surface waters, but numerically dominated the archaeal community in the upper and lower oxycline, and were at low abundances within the OMZ core (Fig. 1C). Both stations display the highest collective archaeal abundances in the surface and upper oxycline with the exception of station 2, where surface water archaeal abundances were relatively low.

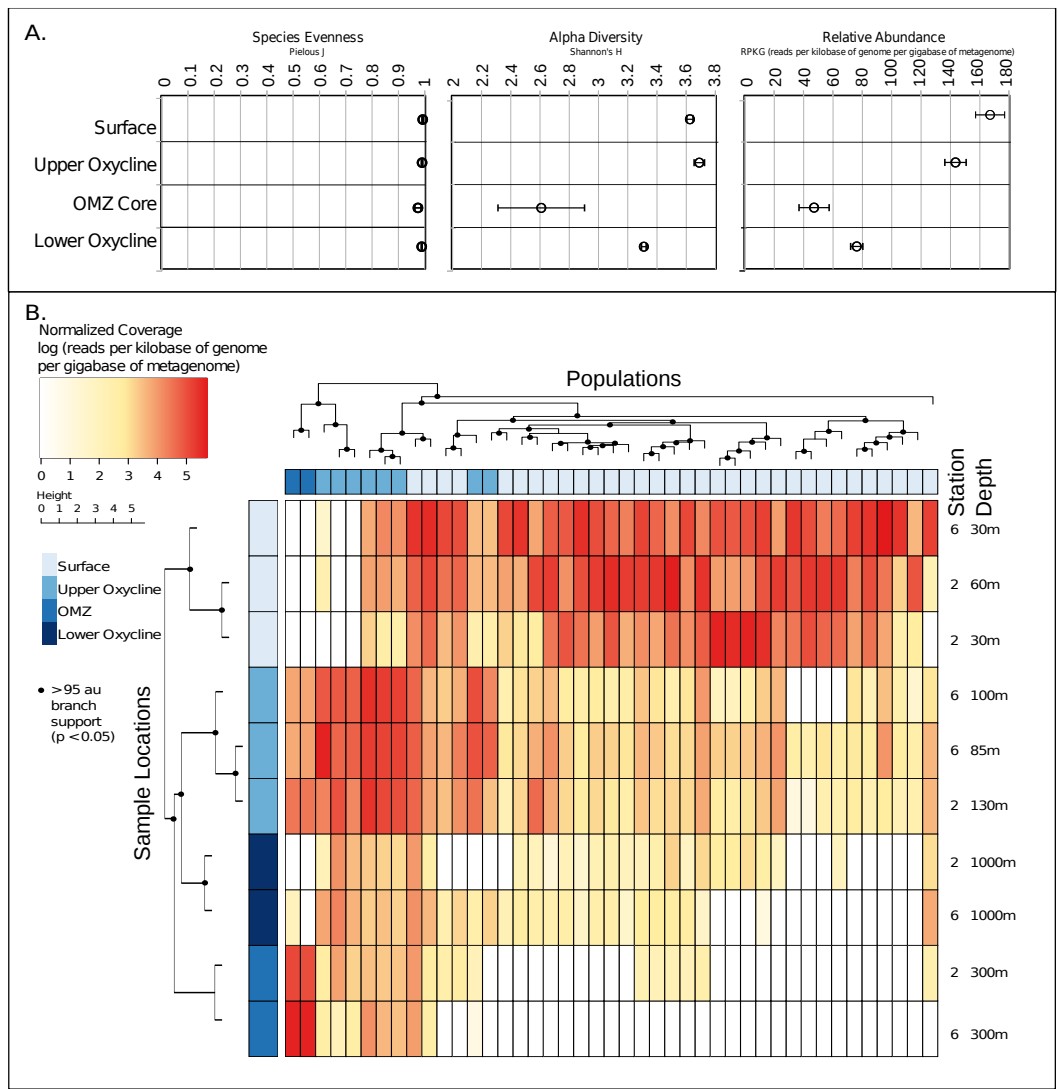

**Figure 4 Ecology and relative abundance of putative archaeal viruses.** (A) Evenness (Pielou's J), alpha diversity (Shannon-Weiner index), and relative abundance, at each sample site and depth. The relative abundance and alpha diversity of putative ETNP archaeal virus populations tracks gradients in oxygen concentration, with the highest abundances found in habitats with high oxygen and vice versa. Viral evenness is high and relatively consistent in each. (B) Relative abundance of archaeal virus populations at each station and sample depth determined by mapping quality trimmed reads to the 43 archaeal virus populations. The units of the abundance measurements are the logarithm (base 10) of the read coverage, normalized by the number of bases in the trimmed read file and contig size. The dendrogram on the $x$-axis represents the 43 ETNP archaeal virus populations, clustered by relative abundance per sample, and the dendrogram on the $y$-axis represents the sample locations grouped according to similarities in contig composition. Nodes with approximately unbiased (AU) bootstrapping values greater then 95 ($p \leq 0.05$) are displayed with a dot.

While other archaeal phyla become enriched at the OMZ core, the lowest total archaeal abundances in either station are found at this depth (Fig. 1C).

That oxygen concentration and temperature in the ETNP appeared to most strongly influence the Archaea and archaeal virus community structure is not surprising given that

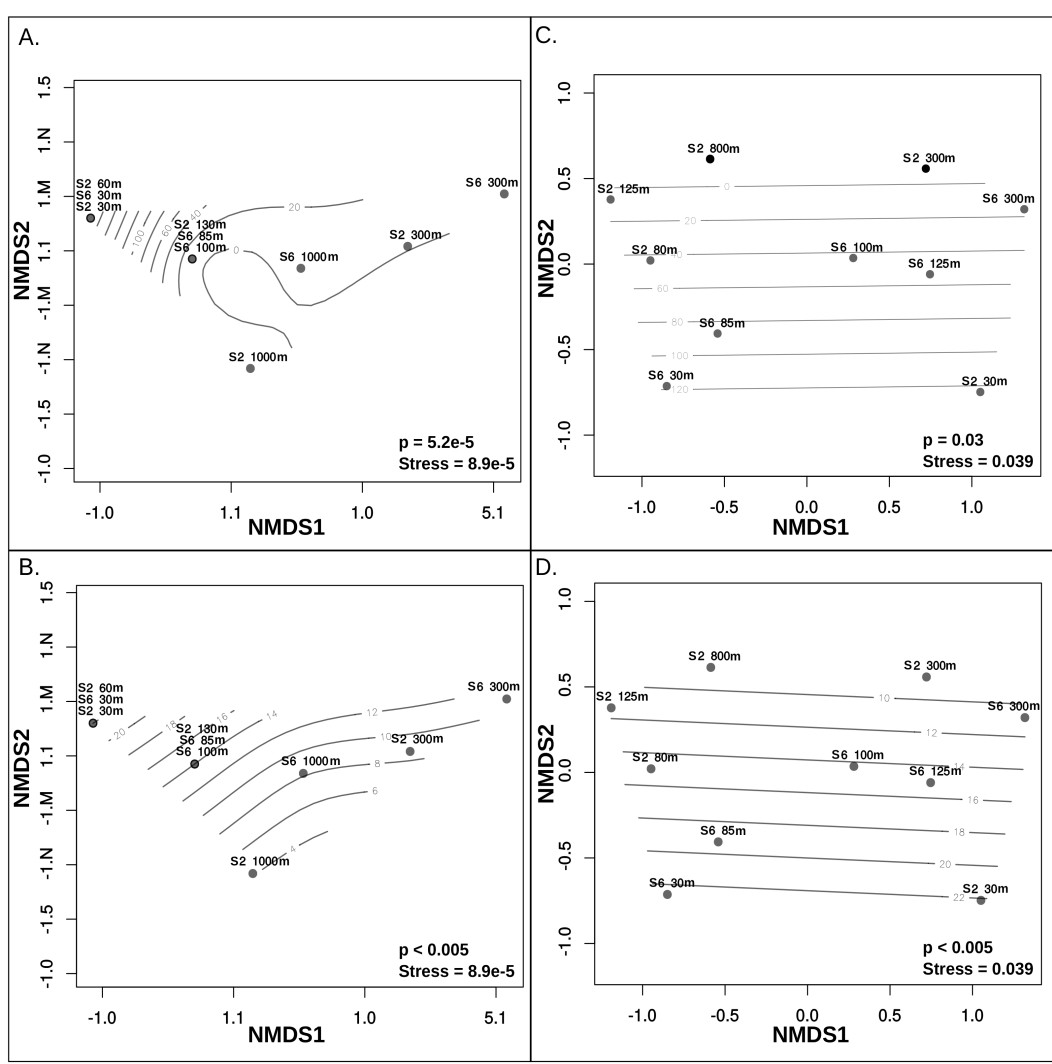

**Figure 5** Non-metric multidimensional scaling analysis of ETNP putative archaeal virus and Archaea population distribution in relation to environmental variables. Measurements of temperature, oxygen, salinity, chlorophyll *a*, and depth were examined to determine which of these features influences the separations among both archaeal virus populations and Archaea in each habitat. Oxygen concentration (A) and Temperature (B) most significantly influenced the ETNP archaeal virus population distribution nonlinearly. Oxygen concentration (C) and Temperature (D) also have a linear impact on the distribution of the Archaea. Depth was also found to significantly influence archaeal virus population distributions but this is likely due to a co-variable effect with oxygen and temperature. Salinity and chlorophyll *a* do not exhibit a significant correlation with the archaeal virus or archaeal population distributions.

numerous studies have revealed the strong effects of oxygen concentration and temperature on microbial community structure in the marine environment, including effects on the archaeal community structure (*Wright, Konwar & Hallam, 2012*; *Allers et al., 2013a*; *Allers et al., 2013b*; *Francis et al., 2005*; *Hawley et al., 2014*). In sunlit, warm, oxygenated waters the Marine Group II *Thermoplasmata* are abundant, representing about 5–35% of total microbial assemblages in a variety of marine environments (*Pernthaler et al., 2002*; *Teira et al., 2004*; *Belmar, Molina & Ulloa, 2011*). These *Euryarchaeota* are thought to influence

global carbon cycling via proteorhodopsin-mediated photoheterotrophy (*Iverson et al., 2012*; *Frigaard et al., 2006*; *Zhang et al., 2015*). At the base of and below the photic zone, microbial respiration consumes oxygen, creating the oxycline and ultimately an OMZ. In this cooler oxycline and OMZ water, *Thaumarchaeota,* formally known as Marine Group I, can represent up to about one-quarter of the total microbial census (*Belmar, Molina & Ulloa, 2011*; *Podlaska et al., 2012*; *Wright, Konwar & Hallam, 2012*) and up to one-fifth of all microbial mRNA transcripts (*Stewart, Ulloa & Delong, 2012*).

The non-linear relationship of both oxygen concentration and temperature on the distribution of the putative archaeal viruses may be reflective of an overarching relationship with the distribution of the host Archaea. Qualitatively, and statistically (though with limited power) the abundance profiles of the 43 archaeal virus populations described here correlate with that of *Thermoplasmatales MGII* (Pearson > 0.8, Spearman > 0.8) (Data S8 and S9) (Fig. S3). This may suggest that some of the 43 putative archaeal viruses infect this archaeal group; however, this conclusion is drawn from limited data and cannot be verified without more direct linkage between virus and host from either more in-depth ordination analysis, matching CRISPR arrays or similar k-mer nucleotide frequencies (*Emerson et al., 2013*; *Roux et al., 2015a*; *Roux et al., 2015b*; *Allers et al., 2013a*; *Allers et al., 2013b*), none of which yielded significant results in this study. Seawater samples could also be examined via microscopy and phageFISH to link viruses to hosts, however this has not been attempted using either Archaea or archaeal viruses. This research suggests that the observed putative archaeal virus populations in the ETNP may infect a small subset of Archaea populations shared between the two divergent archaeal communities observed in both stations.

During the review process for this paper two more articles highlighting marine archaeal viruses were published. In the first *Philosof et al. (2017)* identified 26 putative archaeal virus genomes and genome fragments, which may be associated with the ecologically important *MGII Euryarchaeota*, and appear to encode a unique archaeal replication mechanism. The second describes an additional 58 putative marine archaeal virus genomes derived from the Tara Oceans viromes and Osaka Bay based on homology to Euryarchaeal Marine Group II chaperonin genes phylogenetic analysis using tBLASTx genomic similarity scores (*Nishimura et al., 2017*).

## CONCLUSIONS

Using MArVD, we conservatively identify 43 putative archaeal virus populations from the mesophilic oceans and use these populations to explore archaeal virus ecology in relation to environmental features throughout the ETNP OMZ. We show that the population distribution of these putative archaeal viruses is significantly correlated with the gradient of oxygen concentration and temperature and coincides with the stratification of marine Archaea, suggesting that the observed niche differentiation among these putative archaeal viruses is reflective of the distribution of their potential hosts. We note, however, that our estimates of archaeal viral diversity in this system are likely underestimated, due in part to the still limited representation of confirmed archaeal viruses in public databases. With further advancement of technologies, methods, and reference databases which allow for

the linkage of viruses and hosts in a high throughput manner we can better understand the ecological implications of viral infection, and especially the role of archaeal virus infection, globally.

## ACKNOWLEDGEMENTS

We thank the captain and crew of the R/V New Horizon for sampling opportunity in the ETNP, the Sullivan Lab, the Stewart lab, and the reviewers for vital improvements to the manuscript. We also thank the Ohio Super Computing center for computational support.

### Funding

This work was funded in part by awards from the National Science Foundation (OCE # 1536989) and the Gordon and Betty Moore Foundation (GBMF #3790) to MBS, as well as National Science Foundation (1151698, 1558916, 1564559), Sloan Foundation grant (RC944) and Simons Foundation award (346253) to FJS. There was no additional external funding received for this study. The funders had no role in study design, data collection and analysis, decision to publish, or preparation of the manuscript.

### Grant Disclosures

The following grant information was disclosed by the authors:
National Science Foundation: OCE #1536989.
Gordon and Betty Moore Foundation: GBMF #3790.
National Science Foundation: 1151698, 1558916, 1564559.
Sloan Foundation: RC944.
Simons Foundation: 346253.

### Competing Interests

The authors declare there are no competing interests.

### Author Contributions

- Dean R. Vik Simon Roux, Jennifer R. Brum, Ben Bolduc, Joanne B. Emerson, Cory C. Padilla, Frank J. Stewart and Matthew B. Sullivan conceived and designed the experiments, performed the experiments, analyzed the data, contributed reagents/materials/analysis tools, wrote the paper, prepared figures and/or tables, reviewed drafts of the paper.

### Field Study Permissions

The following information was supplied relating to field study approvals (i.e., approving body and any reference numbers):

Field experiments were approved by the Secretaria de Relaciones (SRE), diplomatic note number CTC/04923/2013, and the Secretariat of Agriculture, Livestock, Rural Development, Fisheries and Food (SAGARPA), number DGOPA.-03597/240413.

## Data Availability

The raw data has been supplied as Supplementary Files. MArVD is available for download and use at (https://bitbucket.org/MAVERICLab/marvd) and on iVirus at the CyVerse Discovery Environment (https://de.cyverse.org/de/). Assembled putative archaeal virus contigs, R scripts and the raw supplemental data are available on iVirus at the CyVerse Discovery Environment at (https://de.cyverse.org/de/) in folder (/iplant/home/shared/iVirus/Vik_et_al_2017_data) and at bitbucket (https://bitbucket.org/MAVERICLab/marvd). Station 2 16S rRNA gene sequences are deposited under bioproject ID: PRJNA345356, Station 6 were previously deposited as Bioproject ID: PRJNA263621.

## Supplemental Information

Supplemental information for this article can be found online at http://dx.doi.org/10.7717/peerj.3428#supplemental-information.

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
