# Peer review of "Putative archaeal viruses from the mesopelagic ocean"

_PeerJ, doi:10.7717/peerj.3428_

## Round 0.1 · original submission · Minor Revisions

Many thanks for your submission to PeerJ. Myself and the reviewers found your article very interesting and very suitable for publication in PeerJ, pending a few minor revisions. I hope this is satisfactory for you.

Many thanks & best wishes,
Dr. Chris Cooper

Reviewer 1 ·

Basic reporting

The manuscript is well written with extensive reference to the published literature. The figures and tables (in the manuscript and in the supplementary data) are of excellent quality and are well described by the legends. I have a few minor comments on typos etc.
a) line 55: 'Resultantly' seems an odd word to me. Perhaps 'Consequently' would be better.
b) line 58: what is 'liable' DOM?
c) line 150: an apostrophe is not needed with bases
d) line 315: do you really mean 'intuitive'?
e) line 333: do you mean rarified or rare/uncommon?
f) please ensure the species names are in italics in the references
g) Fig. 1 legend; the temperatures appear as 28c and 6c in my pdf copy.
h) Fig. 1: I cannot find reference to the chlorophyll units of measurement either in the legend or the materials and methods.

Experimental design

I found the experiments were well designed and executed. The authors explain clearly the need for a rapid means of identifying archaeal viruses in marine metavirome data, and have developed a script to process such data. While this script relies on previous BLAST comparisons and resulting annotations, the python script can be used to speed up the time from data collection to contig analysis, so provides a useful addition to the field.
The authors first test the use of this script on publicly available data, and then apply it to new data (marine viromes from two sites) to identify novel archaeal virus contigs and relate these to the prokaryotic community structure and other environmental parameters at the two marine stations. The methods are well described and references given where needed.

Validity of the findings

The data is rigorously analysed and examined from different perspectives in order to test its robustness. Statistical significance is tested appropriately. I believe the conclusions are supported by the results, and indeed the authors have detected novel virus genomes and linked them to likely archaeal hosts - and so have considerable material for further investigation.

I have a few minor comments:
a) I see that the 16S rRNA sequence data has been submitted to GenBank, but I could not see if the viromes and/or assembled virus contigs would also be submitted?
b) lines 460-464: Given the very low change in salinity, I think this section could be edited down somewhat.
c) How do you intend to distribute your python script, or do you intend to offer a web-based service of some kind? While I have a copy as part of the review process (and I have looked through it!), I am not sure whether it will just be offered as a downloadable file or if you have plans to develop it further as an online resource.

Additional comments

This is a very well designed and executed study with useful results that will be of interest to a wide variety of marine microbiologists/virologists.

Reviewer 2 ·

Basic reporting

The article is well written and reports a novel and potentially important group of archaeal viruses. The quality of the figures could be improved.

Experimental design

The method of assembly of the contigs is not described. A detailed description of how the contigs were generated is of the essence, particularly when assembling metaviriomes that are prone to chimeric assembly.

Validity of the findings

no comment

Additional comments

It is a pity that the genomes are not compared with the ones in Philosof et al that although not officially published yet should be available from databases. At least it would be interesting to compare the genomes at the annotation level. e.g. do the viruses reported here have a replisome like the Philosof et al ones? Some of the novel viruses could be preying on Marine Group III, and this possibility should be mentioned.

---

## Round 0.2 · accepted · Accept

Many thanks for responding to the reviewers comments in detail. Congratulations on your acceptance, and I look forward to seeing the finished article in press!,

Best wishes, Chris